# Active Disturbance Rejection Adaptive Control for Hydraulic Lifting Systems with Valve Dead-Zone

**Fengbo Yang** [1,2]**, Hongping Zhou** [1,2] **and Wenxiang Deng** [3,*]

1   Co-Innovation Center of Efficient Processing and Utilization of Forest Resources, Nanjing Forestry University, Nanjing 210037, China; yfb@njfu.edu.cn (F.Y.); hpzhou@njfu.edu.cn (H.Z.)
2   College of Mechanical and Electronic Engineering, Nanjing Forestry University, Nanjing 210037, China
3   School of Mechanical Engineering, Nanjing University of Science and Technology, Nanjing 210094, China
*   Correspondence: wxdeng@njust.edu.cn; Tel.: +86-025-8431-5125

**Abstract:** In this article, the motion control problem of hydraulic lifting systems subject to parametric uncertainties, unmodeled disturbances, and a valve dead-zone is studied. To surmount the problem, an active disturbance rejection adaptive controller was developed for hydraulic lifting systems. Firstly, the dynamics, including both mechanical dynamics and hydraulic actuator dynamics with a valve dead-zone of the hydraulic lifting system, were modeled. Then, by adopting the system model and a backstepping technique, a composite parameter adaptation law and extended state disturbance observer were successfully combined, which were employed to dispose of the parametric uncertainties and unmodeled disturbances, respectively. This much decreased the learning burden of the extended state disturbance observer, and the high-gain feedback issue could be shunned. An ultimately bounded tracking performance can be assured with the developed control method based on the Lyapunov theory. A simulation example of a hydraulic lifting system was carried out to demonstrate the validity of the proposed controller.

**Keywords:** hydraulic lifting system; disturbance compensation; adaptive control; valve dead-zone

## 1. Introduction

Hydraulic lifting systems are applied widely in modern industry owing to their advantages including large force/torque output, small size-to-power ratio, and high response [1–5]. However, considering that heavy nonlinearities (i.e., friction nonlinearity, transmission nonlinearity, and valve dead-zone) and unmodeled uncertainties (i.e., parametric uncertainties and unmodeled disturbances) exist in hydraulic lifting systems, achieving high-performance motion control for hydraulic lifting systems is still challenging [6–10]. Therefore, to attain the enhancement of tracking accuracy for hydraulic lifting systems, it is essential to study advanced controllers.

Over the past serval decades, many advanced control methods have been developed to obtain high-precision tracking for hydraulic systems. The feedback linearization control in [11,12] was employed to address dynamic nonlinearities. To dispose of parametric uncertainties in hydraulic actuating systems, adaptive control was utilized in [13], while it did nothing about external disturbances [14]. In [14], adaptive robust control was developed to simultaneously handle parametric uncertainties and disturbances and is widely utilized in actual systems [15–21]. Nevertheless, only the boundedness of the tracking error was attained while facing time-varying disturbances. In addition, sliding mode control with a simple structure was employed in [22] to attain anti-disturbance ability for electrohydraulic actuators. An output–feedback-based sliding mode control framework was presented in [23] to obtain the finite-time tracking performance. Nonetheless, as the disturbances increased, the high-gain feedback way in [11–20,22,23] was adopted, which might make the system become uncontrollable.

To eliminate the influence of unmodeled disturbances on tracking performance, many disturbance observers have been proposed. In [24], Yao et al. used an extended state disturbance observer to obtain the estimations of mismatched and matched disturbances. Whereas only the bounded control performance of a hydraulic system can be attained in the face of time-variant disturbances, a new estimator [25] was adopted to deal with unknown uncertainties for servo systems, where the bounded stability was ensured. A novel sliding mode observer in [26] was adopted to handle pressure dynamics and force dynamics in hydraulic actuators, whereas the sign function in the controller could result in chattering and system instability. In [27], Deng et al. developed an extended-state-observer-based adaptive controller to dispose of parameter uncertainties and external disturbances existing in hydraulic systems, in which the system velocity signal was not measured. Moreover, compared with the aforementioned controller methods for hydraulic actuation systems, the controller development of the hydraulic lifting systems presented in this article is more complicated in consideration of its inherent transmission nonlinearities and valve dead-zone. Hence, developing a high-performance controller for a hydraulic lifting system is challenging.

In this article, an active disturbance rejection adaptive control framework was developed for hydraulic lifting systems subject to parametric uncertainties, unmodeled disturbances, and a valve dead-zone. Firstly, the dynamics, including both mechanical dynamics and hydraulic actuator dynamics with a valve dead-zone of the hydraulic lifting system, were modeled. Using the system model and backstepping technique, a composite parameter adaptation law and extended state disturbance observer were successfully combined, which were adopted to dispose of parametric uncertainties and unmodeled disturbances, respectively. This much decreased the learning burden of the extended state disturbance observer, and the high-gain feedback issue could be shunned. In addition, based on the Lyapunov theory, the ultimately bounded tracking performance of the controller could be assured. A simulation example of a hydraulic lifting system was carried out to demonstrate the effectiveness of the proposed controller.

The main contributions of this article are as follows: (1) an active disturbance rejection adaptive control framework was developed for a hydraulic lifting system subject to parametric uncertainties, unmodeled disturbances, and a valve dead-zone, in which the ultimately bounded tracking performance of the controller could be assured; (2) by adopting the adaptive control, the learning burden of the extended state disturbance observer could be much reduced; therefore, the high-gain feedback way could be shunned; (3) the merits of the developed control method were verified by simulation results.

The structure of this paper is as follows: a system description of the hydraulic lifting system is shown in Section 2. The controller design and its stability proof can be found in Section 3. The simulation results and conclusions are presented in Sections 4 and 5, respectively.

## 2. System Description

The hydraulic lifting system under study is presented in Figure 1. As shown in Figure 1, $O$ stands for the gyration center; $O_1$ and $O_2$ denote the rotation centers of the upper and lower ears of the hydraulic cylinder, respectively; $O_3$ stands for the lifting system barycenter; $q$ stands for the lifting arm rotary angle; $G$ and $F$ denote the force acting on the arm and the arm gravity, respectively. In addition, let $OO_2 = L_1$, $O_1O_2 = L_2$, $OO_1 = L_3$, $OO_3 = L_4$, $O_1'O_2 = L$, $\angle O_1OO_1' = q$, $\angle O_1OO_2 = q_0$, $\angle O_1OO_3 = \beta_0$, and $\angle OO_1'O_2 = \alpha$.

The dynamics of the lifting system can be written as:

$$J\ddot{q} = \left(\frac{\partial x_p}{\partial q}\right)(P_1 A_1 - P_2 A_2) - mgL_4\cos(q + \beta_0) - B\dot{q} - A_f S_f(\dot{q}) + d_1(t) \qquad (1)$$

where $J$ and $m$ stand for the rotary inertia and arm mass, respectively; $P_1$ and $P_2$ stand for the pressure values of two chambers of the hydraulic cylinder, respectively; $A_1$ and $A_2$ stand for the effective areas of two chambers in the hydraulic cylinder, respectively; $A_f$ and $S_f$ stand

for the amplitude and approximated shape function of the Coulomb friction, respectively; $B$ stands for the viscous friction coefficient; $d_1(t)$ stands for the unmodeled disturbances.

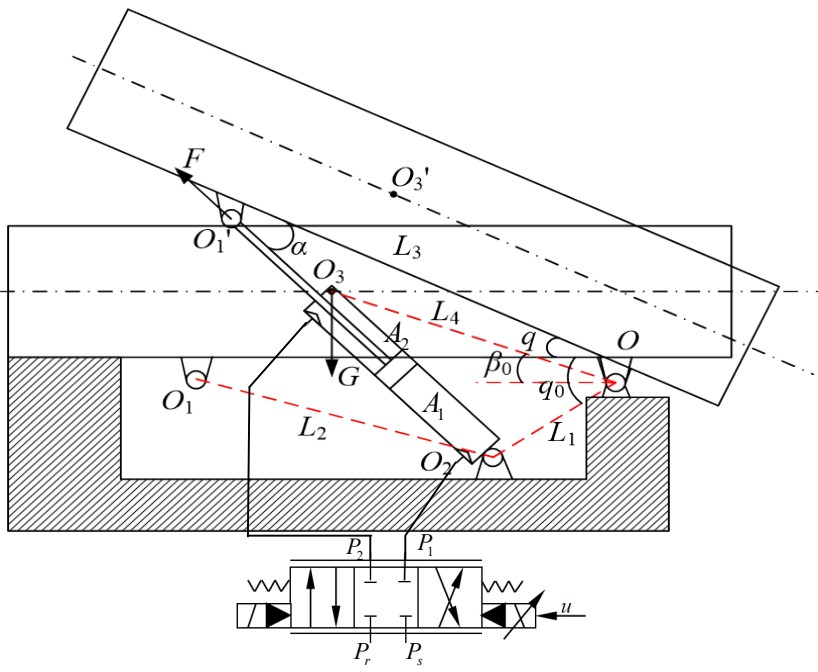

**Figure 1.** A sketch of the lifting system.

Defining the cylinder displacement as $x_p$ obtains:

$$
\begin{aligned}
x_p &= L - L_2 \\
&= \sqrt{L_1^2 + L_3^2 - 2L_1L_3\cos(q + q_0)} - L_2
\end{aligned}
\tag{2}
$$

There exists:

$$
\frac{\partial x_p}{\partial q} = \frac{L_1L_3\sin(q + q_0)}{\sqrt{L_1^2 + L_3^2 - 2L_1L_3\cos(q + q_0)}}
\tag{3}
$$

Considering the oil compressibility, the pressure dynamics of the hydraulic actuator are written as [28]:

$$
\begin{aligned}
\dot{P}_1 &= \frac{\beta_e}{V_1(q)}\left[-A_1\frac{\partial x_p}{\partial q}\dot{q} - C_t(P_1 - P_2) + Q_1 + d_{21}(t)\right] \\
\dot{P}_2 &= \frac{\beta_e}{V_2(q)}\left[A_2\frac{\partial x_p}{\partial q}\dot{q} + C_t(P_1 - P_2) - Q_2 - d_{22}(t)\right]
\end{aligned}
\tag{4}
$$

where $V_1 = V_{01} + A_1x_s$ and $V_2 = V_{02} - A_2x_s$ stand for the volumes of the two chambers, respectively; $V_{01}$ and $V_{02}$ stand for the original volumes of the two chambers, respectively; $\beta_e$ stands for the effective oil bulk modulus; $C_t$ stands for the internal leakage coefficient; $Q_1$ and $Q_2$ stand for the return flow and supplied flow of the hydraulic cylinder, respectively; $d_{21}(t)$ and $d_{22}(t)$ stand for the unmodeled disturbances.

Consequently, the flows $Q_1$ and $Q_2$ thus are modeled as [28]:

$$
\begin{aligned}
Q_1 &= k_q x_v[s(x_v)\sqrt{P_s - P_1} + s(-x_v)\sqrt{P_1 - P_r}] \\
Q_2 &= k_q x_v[s(x_v)\sqrt{P_2 - P_r} + s(-x_v)\sqrt{P_s - P_2}]
\end{aligned}
\tag{5}
$$

where $k_q$ stands for the flow gain; $x_v$ stands for the valve spool displacement; $P_r$ and $P_s$ stand for the return pressure and supply pressure, respectively; $s(*)$ is defined as:

$$s(*) = \begin{cases} 1, & \text{if } * \geq 0 \\ 0, & \text{if } * < 0 \end{cases} \tag{6}$$

Given that the compressibility flow is small and the valve window is configured and symmetrical, one obtains [28]:

$$P_s \approx P_1 + P_2 \tag{7}$$

Defining the load pressure $P_L = P_1 - \overline{A}_c P_2$, where $\overline{A}_c = A_2 / A_1$, it is easy to derive:

$$P_1 = \frac{\overline{A}_c P_s + P_L}{1 + \overline{A}_c}, \; P_2 = \frac{P_s - P_L}{1 + \overline{A}_c} \tag{8}$$

Defining the state variables $x = [x_1; x_2; x_3] = [q; \dot{q}; A_1 P_L / J]$, the system's state-space form using (1)–(5) and (8) is:

$$\dot{x}_1 = x_2$$
$$\dot{x}_2 = \frac{\partial x_p}{\partial x_1} x_3 - \frac{m}{J} g_1(x_1) - \frac{B}{J} x_2 - \frac{A_f}{J} S_f(x_2) + \Delta_1(t) \tag{9}$$
$$\dot{x}_3 = \frac{\beta_e k_t}{J} g_2(x_v, x_3) \cdot x_v(u) - \frac{\beta_e}{J} g_3(x_1, x_2) - \frac{\beta_e C_t}{J} g_4(x_1, x_3) + \Delta_2(t)$$

where

$$g_1(x_1) = g L_4 cos(x_1 + \beta_0)$$
$$\Delta_1(t) = d_1(t) / J \tag{10}$$
$$\Delta_2(t) = \beta_e (A_1 d_{21}(t) / V_1 + A_2 d_{22}(t) / V_2) / J$$

and

$$g_2(x_v, x_3) = \frac{A_1 R_1'}{V_1} + \frac{A_2 R_2'}{V_2},$$
$$R_1' = s(x_v) \sqrt{P_s - \frac{\overline{A}_c P_s + J x_3 / A_1}{1 + \overline{A}_c}} + s(-x_v) \sqrt{\frac{\overline{A}_c P_s + J x_3 / A_1}{1 + \overline{A}_c} - P_r},$$
$$R_2' = s(x_v) \sqrt{\frac{P_s - J x_3 / A_1}{1 + \overline{A}_c} - P_r} + s(-x_v) \sqrt{P_s - \frac{P_s - J x_3 / A_1}{1 + \overline{A}_c}}, \tag{11}$$
$$g_3(x_1, x_2) = \frac{\partial x_p}{\partial x_1} \left(\frac{A_1^2}{V_1} + \frac{A_2^2}{V_2}\right) x_2, \; g_4(x_1, x_3) = \left(\frac{A_1}{V_1} + \frac{A_2}{V_2}\right) \frac{(\overline{A}_c - 1) P_s + \frac{2 J x_3}{A_1}}{1 + \overline{A}_c}$$

Omitting the valve dynamics, the valve spool displacement, $x_v$, can be modeled as a static mapping of the control input voltage, $u$, with a dead-zone [29]

$$x_v(u) = DZ(u) = \begin{cases} m_r u - m_r b_r, & \text{if } u \geq b_r \\ 0, & \text{if } b_l < u < b_r \\ m_l u - m_l b_l, & \text{if } u \leq b_l \end{cases} \tag{12}$$

where $DZ(u)$ stands for the valve dead-zone nonlinear function; $m_r > 0$, $m_l > 0$, $b_r \geq 0$, and $b_l \geq 0$ denote the unknown constants of the right slope, left slope, right break-point, and left break-point of the dead-zone, respectively. The characteristic of the valve dead-zone is given in Figure 2.

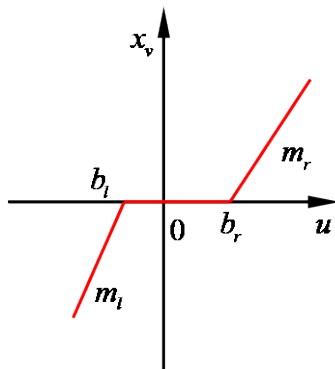

**Figure 2.** The valve dead-zone characteristics.

**Assumption 1.** *Although the valve dead-zone parameters are unknown, the maximum and minimum values of the left slope and right slope are known, i.e.,* $\overline{m} = max\{m_r, m_l\}$ *and* $\underline{m} = min\{m_r, m_l\}$, *where* $\overline{m}$ *and* $\underline{m}$ *are known positive constants.*

Based on Assumption 1, Formula (12) can be reconstructed as:

$$x_v(u) = m(t)u + d(t) \tag{13}$$

in which

$$m(t) \triangleq \begin{cases} m_l, & \text{if } u \leq 0 \\ m_r, & \text{if } u > 0 \end{cases} \tag{14}$$

and

$$d(t) \triangleq \begin{cases} -m_r b_r, & \text{if } u \geq b_r \\ -m(t)u, & \text{if } b_l < u < b_r \\ -m_l b_l, & \text{if } u \leq b_l \end{cases} \tag{15}$$

It is worth noting that:

$$\frac{m(t)}{\underline{m}} = 1 + k(t) \tag{16}$$

where $k(t)$ stands for a positive bounded piecewise continuous function.

Taking the parametric uncertainties into consideration, defining the unknown parameter vector, $\theta = [\theta_1; \theta_2; \theta_3; \theta_4; \theta_5; \theta_6] = \left[ \frac{m}{J}; \frac{B}{J}; \frac{A_f}{J}; \frac{\beta_e k_t}{J}; \frac{\beta_e}{J}; \frac{\beta_e C_t}{J} \right]$. According to (9), (13) and (16) can be rewritten by:

$$\begin{aligned} \dot{x}_1 &= x_2 \\ \dot{x}_2 &= \frac{\partial x_p}{\partial x_1} x_3 - \theta_1 g_1(x_1) - \theta_2 x_2 - \theta_3 S_f(x_2) + \Delta_1(t) \\ \dot{x}_3 &= \theta_4 g_2(x_v, x_3)u - \theta_5 g_3(x_1, x_2) - \theta_6 g_4(x_1, x_3) + \overline{\Delta}_2(t) \end{aligned} \tag{17}$$

in which $\overline{\Delta}_2(t) = \Delta_2(t) + \underline{m}k(t)u + d(t)$.

Before developing the controller, the following assumptions were provided:

**Assumption 2.** *The desired trajectory* $x_{1d} \in \mathbb{C}^3$ *is bounded.*

**Assumption 3.** $\Delta_1(t)$ *and* $\overline{\Delta}_2(t)$ *in (17) meet:*

$$|\Delta_1(t)| \leq \delta_1, \ \left| \overline{\Delta}_2(t) \right| \leq \delta_2 \tag{18}$$

*where* $\delta_i$ *(i = 1, 2) are unknown positive constants. In addition, the set of parameters* $\theta$ *satisfies:*

$$\theta \in \Omega_\theta \triangleq \{\theta : \theta_{min} \leq \theta \leq \theta_{max}\} \tag{19}$$

*where* $\theta_{max} = [\theta_{1max}; \theta_{2max}; \theta_{3max}; \theta_{4max}; \theta_{5max}; \theta_{6max}]$, $\theta_{min} = [\theta_{1min}; \theta_{2min}; \theta_{3min}; \theta_{4min}; \theta_{5min}; \theta_{6min}]$.

## 3. Controller Design

*3.1. Discontinuous Mapping and Parameter Adaptive Law*

Defining the estimation values of $\theta$ as $\hat{\theta}$, and the estimation error $\widetilde{\theta} = \hat{\theta} - \theta$, the discontinuous mapping was designed as [30]:

$$\text{Proj}_{\hat{\theta}_i}(\bullet_i) = \begin{cases} 0, & \text{if } \hat{\theta}_i = \theta_{i\max} \text{ and } \bullet_i > 0 \\ 0, & \text{if } \hat{\theta}_i = \theta_{i\min} \text{ and } \bullet_i < 0 \\ \bullet_i, & \text{otherwise} \end{cases} \tag{20}$$

in which $i = 1, 2, 3, 4, 5$, and 6. The adopted updated law is set as:

$$\dot{\hat{\theta}} = \text{Proj}_{\hat{\theta}}(\mathbf{\Gamma}\boldsymbol{\tau})\ \hat{\theta}(0) \in \Omega_{\boldsymbol{\theta}} \tag{21}$$

where $\text{Proj}_{\hat{\theta}}(\bullet) = [\text{Proj}_{\hat{\theta}_1}(\bullet_1), \ldots, \text{Proj}_{\hat{\theta}_6}(\bullet_6)]^T$; $\mathbf{\Gamma} \in \mathbb{R}^{6 \times 6}$ denotes a positive diagonal matrix, for any adapted function $\boldsymbol{\tau}$, the projection mapping (20) ensures [30]:

$$\hat{\boldsymbol{\theta}} \in \Omega_{\hat{\boldsymbol{\theta}}} = \left\{ \hat{\boldsymbol{\theta}} : \boldsymbol{\theta}_{\min} \leq \hat{\boldsymbol{\theta}} \leq \boldsymbol{\theta}_{\max} \right\} \tag{22}$$

$$\widetilde{\boldsymbol{\theta}}^T[\mathbf{\Gamma}^{-1}\text{Proj}_{\hat{\boldsymbol{\theta}}}(\mathbf{\Gamma}\boldsymbol{\tau}) - \boldsymbol{\tau}] \leq 0, \quad \forall \boldsymbol{\tau} \tag{23}$$

*3.2. Disturbance Observer Design*

To realize the observation of matching and mismatching disturbances in (17), the disturbance observer should be designed first. We defined the disturbances $\Delta_1(t)$ and $\overline{\Delta}_2(t)$ as the extended states $x_{e1}$ and $x_{e2}$, and the variables $H_1(t) = \dot{x}_{ei}$ for $i = 1, 2$. In addition, we assumed the functions $H_1(t)$ and $H_2(t)$ were unknown bounded. Hence, based on (17), two extended state disturbance observers were constructed as:

$$\begin{cases} \dot{\hat{x}}_1 = \hat{x}_2 + 3\omega_{o1}(x_1 - \hat{x}_1) \\ \dot{\hat{x}}_2 = \frac{\partial x_p}{\partial x_1}x_3 + \hat{\theta}^T\boldsymbol{\varphi}_2 + \hat{x}_{e1} + 3\omega_{o1}^2(x_1 - \hat{x}_1) \\ \dot{\hat{x}}_{e1} = \omega_{o1}^3(x_1 - \hat{x}_1) \end{cases} \tag{24}$$

$$\begin{cases} \dot{\hat{x}}_3 = \hat{\theta}^T\boldsymbol{\varphi}_3 + \hat{x}_{e2} + 2\omega_{o2}(x_3 - \hat{x}_3) \\ \dot{\hat{x}}_{e2} = \omega_{o2}^2(x_3 - \hat{x}_3) \end{cases} \tag{25}$$

where $\boldsymbol{\varphi}_2 = \left[-g_1(x_1); -x_2; -S_f(x_2); 0; 0; 0\right]$; $\boldsymbol{\varphi}_3 = [0; 0; 0; g_2(x_v, x_3)u; -g_3(x_1, x_2); -g_4(x_1, x_3)]$; $\omega_{o1}$ and $\omega_{o2}$ denote the adjustable parameters of observers (24) and (25); $\hat{\cdot}$ denotes the estimation of $\cdot$, and the estimation error of $\cdot$ is defined as $\widetilde{\cdot} = \hat{\cdot} - \cdot$ throughout the paper.

Considering the existence of the parametric uncertainties, $\widetilde{\theta}$, and the disturbances, $\Delta_1(t)$ and $\overline{\Delta}_2(t)$, the following two constructed forms of $x_{e1}$ and $x_{e2}$ are given as:

(1)  Define $x_{e1} = \Delta_1(t)$ and $x_{e2} = \overline{\Delta}_2(t)$; then, (17) is rewritten as:

$$\begin{cases} \dot{x}_1 = x_2 \\ \dot{x}_2 = \frac{\partial x_p}{\partial x_1}x_3 + \hat{\theta}^T\boldsymbol{\varphi}_2 - \widetilde{\theta}^T\boldsymbol{\varphi}_2 + x_{e1} \\ \dot{x}_{e1} = H_1(t) \end{cases} \tag{26}$$

$$\begin{cases} \dot{x}_3 = \hat{\theta}^T\boldsymbol{\varphi}_3 - \widetilde{\theta}^T\boldsymbol{\varphi}_3 + x_{e2} \\ \dot{x}_{e2} = H_2(t) \end{cases} \tag{27}$$

Defining $\eta = [\eta_1, \eta_2, \eta_3]^T = [\widetilde{x}_1, \widetilde{x}_2/\omega_{o1}, \widetilde{x}_{e1}/\omega_{o1}^2]^T$, $\chi = [\chi_1, \chi_2]^T = [\widetilde{x}_3, \widetilde{x}_{e2}/\omega_{o2}]^T$, and based on (24)–(27), the dynamics of the designed observers are presented as:

$$\begin{cases} \dot{\eta} = \omega_{o1}A_1\eta - B_1\frac{\widetilde{\theta}^T\boldsymbol{\varphi}_2}{\omega_{o1}} + B_2\frac{H_1(t)}{\omega_{o1}^2} \\ \dot{\chi} = \omega_{o2}A_2\chi - B_3\widetilde{\theta}^T\boldsymbol{\varphi}_3 + B_4\frac{H_2(t)}{\omega_{o2}} \end{cases} \tag{28}$$

in which $B_1 = [0, 1, 0]^T$, $B_2 = [0, 0, 1]^T$, $B_3 = [1, 0]^T$, $B_4 = [0, 1]^T$ and

$$A_1 = \begin{bmatrix} -3 & 1 & 0 \\ -3 & 0 & 1 \\ -1 & 0 & 0 \end{bmatrix}, A_2 = \begin{bmatrix} -2 & 1 \\ -1 & 0 \end{bmatrix} \circ \qquad (29)$$

(2) Define $x_{e1} = \Delta_1(t) - \widetilde{\boldsymbol{\theta}}^T \boldsymbol{\varphi}_2$ and $x_{e2} = \overline{\Delta}_2(t) - \widetilde{\boldsymbol{\theta}}^T \boldsymbol{\varphi}_3$; then, (17) is rewritten as:

$$\begin{cases} \dot{x}_1 = x_2 \\ \dot{x}_2 = \frac{\partial x_p}{\partial x_1} x_3 + \hat{\boldsymbol{\theta}}^T \boldsymbol{\varphi}_2 + x_{e1} \\ \dot{x}_{e1} = H_1(t) \end{cases} \qquad (30)$$

$$\begin{cases} \dot{x}_3 = \hat{\boldsymbol{\theta}}^T \boldsymbol{\varphi}_3 + x_{e2} \\ \dot{x}_{e2} = H_2(t) \end{cases} \qquad (31)$$

Based on (24), (25), (30) and (31), the dynamics of the designed observers are presented as:

$$\begin{cases} \dot{\boldsymbol{\eta}} = \omega_{o1} A_1 \boldsymbol{\eta} + B_2 \frac{H_1(t)}{\omega_{o1}^2} \\ \dot{\boldsymbol{\chi}} = \omega_{o2} A_2 \boldsymbol{\chi} + B_4 \frac{H_2(t)}{\omega_{o2}} \end{cases} \qquad (32)$$

Since the matrices $A_1$ and $A_2$ are Hurwitz, there always exist positive definite matrices $P_1$ and $P_2$ meeting $A_1^T P_1 + P_1 A_1 = -I$, and $A_2^T P_2 + P_2 A_2 = -I$, where $I$ denotes the unit matrix. Hence, one has:

$$P_1 = \begin{bmatrix} 1 & -\frac{1}{2} & -1 \\ -\frac{1}{2} & 1 & -\frac{1}{2} \\ -1 & -\frac{1}{2} & 4 \end{bmatrix}, P_2 = \begin{bmatrix} \frac{1}{2} & -\frac{1}{2} \\ -\frac{1}{2} & \frac{3}{2} \end{bmatrix} \circ \qquad (33)$$

### 3.3. Controller Design

In this section, the backstepping method [31] was utilized to develop the controller. The details are as follows:

Step 1: Several error variables were designed as below:

$$\begin{aligned} z_2 &= \dot{z}_1 + k_1 z_1 = x_2 - \alpha_1, \\ \alpha_1 &= \dot{x}_{1d} - k_1 z_1, \ z_3 = x_3 - \alpha_2 \end{aligned} \qquad (34)$$

where $z_1 = x_1 - x_{1d}$ denotes the tracking error; $k_1$ stands for the feedback gain.

According to (17) and (34), and differentiating $z_2$, there is:

$$\begin{aligned} \dot{z}_2 &= \dot{x}_2 - \dot{\alpha}_1 \\ &= \frac{\partial x_p}{\partial x_1}(z_3 + \alpha_2) + \hat{\boldsymbol{\theta}}^T \boldsymbol{\varphi}_2 - \widetilde{\boldsymbol{\theta}}^T \boldsymbol{\varphi}_2 + \Delta_1(t) - \dot{\alpha}_1 \end{aligned} \qquad (35)$$

From (35), the virtual control $\alpha_2$ is constructed as:

$$\begin{aligned} \alpha_2 &= \alpha_{2a} + \alpha_{2s}, \\ \alpha_{2a} &= \left(\frac{\partial x_p}{\partial x_1}\right)^{-1}(-\hat{\boldsymbol{\theta}}^T \boldsymbol{\varphi}_2 + \dot{\alpha}_1 - \hat{x}_{e1}), \\ \alpha_{2s} &= -\left(\frac{\partial x_p}{\partial x_1}\right)^{-1} k_2 z_2 \end{aligned} \qquad (36)$$

where $k_2$ stands for the feedback gain.

Putting (36) into (35), one obtains:

$$\dot{z}_2 = \frac{\partial x_p}{\partial x_1} z_3 - k_2 z_2 + \widetilde{x}_{e1} + \Delta_1(t) - \widetilde{\boldsymbol{\theta}}^T \boldsymbol{\varphi}_2 - x_{e1} \qquad (37)$$

Step 2: On the basis of (17) and (34), and differentiating $z_3$, there is:

$$\begin{aligned}\dot{z}_3 &= \dot{x}_3 - \dot{\alpha}_2 \\ &= \hat{\theta}_4 g_2(x_v, x_3)u - \hat{\theta}_5 g_3(x_1, x_2) - \hat{\theta}_6 g_4(x_1, x_3) - \tilde{\boldsymbol{\theta}}^T \boldsymbol{\varphi}_3 + \overline{\Delta}_2(t) - \dot{\alpha}_2\end{aligned} \tag{38}$$

Given that $\dot{\alpha}_2$ contains the incalculable part, it has:

$$\begin{aligned}\dot{\alpha}_2(t, x_1, x_2, \hat{\boldsymbol{\theta}}, \hat{x}_{e1}) &= \dot{\alpha}_{2c} + \dot{\alpha}_{2u}, \\ \dot{\alpha}_{2c} &= \frac{\partial \alpha_2}{\partial t} + \frac{\partial \alpha_2}{\partial x_1}x_2 + \frac{\partial \alpha_2}{\partial x_2}\left(\frac{\partial x_p}{\partial x_1}x_3 + \hat{\boldsymbol{\theta}}^T\boldsymbol{\varphi}_2 + \hat{x}_{e1}\right) + \frac{\partial \alpha_2}{\partial \hat{\theta}}\dot{\hat{\boldsymbol{\theta}}} + \frac{\partial \alpha_2}{\partial x_{e1}}\dot{\hat{x}}_{e1}, \\ \dot{\alpha}_{2u} &= \frac{\partial \alpha_2}{\partial x_2}\left(\Delta_1(t) - \tilde{\boldsymbol{\theta}}^T\boldsymbol{\varphi}_2 - x_{e1} + \tilde{x}_{e1}\right)\end{aligned} \tag{39}$$

in which $\dot{\alpha}_{2c}$ stands for the calculable part; $\dot{\alpha}_{2u}$ stands for the incalculable part.

Hence, the controller input, $u$, is designed as below:

$$\begin{aligned}u &= u_a + u_s, \\ u_a &= \frac{1}{\hat{\theta}_4 g_2(x_v, x_3)}[\hat{\theta}_5 g_3(x_1, x_2) + \hat{\theta}_6 g_4(x_1, x_3) + \dot{\alpha}_{2c} - \hat{x}_{e2}], \\ u_s &= -\frac{1}{\hat{\theta}_4 g_2(x_v, x_3)}k_3 z_3\end{aligned} \tag{40}$$

where $k_3$ stands for the feedback gain.

Putting (40) into (38), one obtains:

$$\dot{z}_3 = -k_3 z_3 + \tilde{x}_{e2} + \overline{\Delta}_2(t) - \tilde{\boldsymbol{\theta}}^T\boldsymbol{\varphi}_3 - x_{e2} - \dot{\alpha}_{2u} \tag{41}$$

### 3.4. Stability Analysis

The main results of this paper are provided below.

**Theorem 1.** *If the uncertain nonlinearities are constants (i.e.,), with Assumptions 1–3, the control law (40), choosing the adaption function $\boldsymbol{\tau}$ as:*

$$\boldsymbol{\tau} = a_2\boldsymbol{\varphi}_2 z_2 + a_3\left(\boldsymbol{\varphi}_3 - \frac{\partial \alpha_2}{\partial x_2}\boldsymbol{\varphi}_2\right)z_3 + c_2\boldsymbol{\eta}^T\boldsymbol{P}_1\boldsymbol{B}_1\frac{\boldsymbol{\varphi}_2}{\omega_{o1}} + c_3\boldsymbol{\chi}^T\boldsymbol{P}_2\boldsymbol{B}_3\boldsymbol{\varphi}_3 \tag{42}$$

*and picking up the proper control parameters, $k_1$, $k_2$, and $k_3$, and the positive coordinating parameters $a_j$ and $c_j$ for $j = 2, 3$, such that the matrix $\Lambda$ defined below is positive definite:*

$$\boldsymbol{\Lambda} = \begin{bmatrix} \boldsymbol{\Lambda}_1 & 0 & \boldsymbol{\Lambda}_3 \\ 0 & \wp_1 & 0 \\ \boldsymbol{\Lambda}_3^T & 0 & \boldsymbol{\Lambda}_2 \end{bmatrix} \tag{43}$$

*in which 0 denotes the zero vector and*

$$\boldsymbol{\Lambda}_1 = \begin{bmatrix} k_1 & -\frac{1}{2} & 0 \\ -\frac{1}{2} & k_2 a_2 & -\frac{a_2}{2}\frac{\partial x_p}{\partial x_1} \\ 0 & -\frac{a_2}{2}\frac{\partial x_p}{\partial x_1} & k_3 a_3 \end{bmatrix}, \boldsymbol{\Lambda}_2 = \begin{bmatrix} \wp_1 & 0 & 0 & 0 \\ 0 & \wp_1 & 0 & 0 \\ 0 & 0 & \wp_2 & 0 \\ 0 & 0 & 0 & \wp_2 \end{bmatrix}, \boldsymbol{\Lambda}_3 = \begin{bmatrix} 0 & 0 & 0 & 0 \\ 0 & -\frac{\gamma_1}{2} & 0 & 0 \\ 0 & -\frac{\gamma_2}{2} & 0 & -\frac{\gamma_3}{2} \end{bmatrix} \tag{44}$$

*in which*

$$\begin{aligned}\wp_1 &= \frac{c_2(\omega_{o1}-1)}{2}, \ \wp_2 = \frac{c_3(\omega_{o2}-1)}{2}, \\ \gamma_1 &= a_2\omega_{o1}^2, \ \gamma_2 = a_3\omega_{o1}^2\left|\frac{\partial \alpha_2}{\partial x_2}\right|, \ \gamma_3 = a_3\omega_{o2}\end{aligned} \tag{45}$$

*then it can be concluded that all the closed-loop signals are bounded, and asymptotic tracking performance is also realized, i.e., $z_1 \to 0$ as $t \to +\infty$.*

**Proof of Theorem 1.** For this case, define $x_{e1} = \Delta_1(t)$ and $x_{e2} = \overline{\Delta}_2(t)$, and choose a Lyapunov function as:

$$V_1 = \frac{1}{2}z_1^2 + \frac{1}{2}a_2z_2^2 + \frac{1}{2}a_3z_3^2 + \frac{1}{2}c_2\boldsymbol{\eta}^T\boldsymbol{P}_1\boldsymbol{\eta} + \frac{1}{2}c_3\boldsymbol{\chi}^T\boldsymbol{P}_2\boldsymbol{\chi} + \frac{1}{2}\widetilde{\boldsymbol{\theta}}^T\boldsymbol{\Gamma}^{-1}\widetilde{\boldsymbol{\theta}} \tag{46}$$

According to (28), (34), (37) and (41), there is:

$$\begin{aligned}
\dot{V}_1 &= z_1\dot{z}_1 + a_2z_2\dot{z}_2 + a_3z_3\dot{z}_3 + \frac{1}{2}c_2\dot{\boldsymbol{\eta}}^T\boldsymbol{P}_1\boldsymbol{\eta} + \frac{1}{2}c_2\boldsymbol{\eta}^T\boldsymbol{P}_1\dot{\boldsymbol{\eta}} \\
&\quad + \frac{1}{2}c_3\dot{\boldsymbol{\chi}}^T\boldsymbol{P}_2\boldsymbol{\chi} + \frac{1}{2}c_3\boldsymbol{\chi}^T\boldsymbol{P}_2\dot{\boldsymbol{\chi}} + \widetilde{\boldsymbol{\theta}}^T\boldsymbol{\Gamma}^{-1}\dot{\hat{\boldsymbol{\theta}}} \\
&= z_1(z_2 - k_1z_1) + a_2z_2(\frac{\partial x_p}{\partial x_1}z_3 - k_2z_2 + \widetilde{x}_{e1} - \widetilde{\boldsymbol{\theta}}^T\boldsymbol{\varphi}_2) + a_3z_3[-k_3z_3 + \widetilde{x}_{e2} \\
&\quad - \widetilde{\boldsymbol{\theta}}^T\boldsymbol{\varphi}_3 - \frac{\partial \alpha_2}{\partial x_2}(-\widetilde{\boldsymbol{\theta}}^T\boldsymbol{\varphi}_2 + \widetilde{x}_{e1})] - \frac{1}{2}c_2\omega_{o1}\|\boldsymbol{\eta}\|^2 - c_2\boldsymbol{\eta}^T\boldsymbol{P}_1\boldsymbol{B}_1\frac{\widetilde{\boldsymbol{\theta}}^T\boldsymbol{\varphi}_2}{\omega_{o1}} \\
&\quad - \frac{1}{2}c_3\omega_{o2}\|\boldsymbol{\chi}\|^2 - c_3\boldsymbol{\chi}^T\boldsymbol{P}_2\boldsymbol{B}_3\widetilde{\boldsymbol{\theta}}^T\boldsymbol{\varphi}_3 + \widetilde{\boldsymbol{\theta}}^T\boldsymbol{\Gamma}^{-1}\dot{\hat{\boldsymbol{\theta}}}
\end{aligned} \tag{47}$$

Putting (42) into (47), it has:

$$\begin{aligned}
\dot{V}_1 &\leq -k_1z_1^2 + |z_1||z_2| + a_2\frac{\partial x_p}{\partial x_1}|z_2||z_3| - k_2a_2z_2^2 + a_2\omega_{o1}^2|z_2||\eta_3| \\
&\quad - k_3a_3z_3^2 + a_3\omega_{o2}|z_3||\chi_2| + a_3\omega_{o1}^2\left|\frac{\partial \alpha_2}{\partial x_2}\right||z_3||\eta_3| - \frac{1}{2}c_2(\omega_{o1}-1)\|\boldsymbol{\eta}\|^2 \\
&\quad - \frac{1}{2}c_3(\omega_{o2}-1)\|\boldsymbol{\chi}\|^2 - \frac{1}{2}c_2\|\boldsymbol{\eta}\|^2 - \frac{1}{2}c_3\|\boldsymbol{\chi}\|^2
\end{aligned} \tag{48}$$

This results in:

$$\begin{aligned}
\dot{V}_1 &\leq -\boldsymbol{Z}^T\boldsymbol{\Lambda}\boldsymbol{Z} \\
&\leq -\lambda_{\min}(\boldsymbol{\Lambda})(\boldsymbol{z}^T\boldsymbol{z} + \boldsymbol{\eta}_a^T\boldsymbol{\eta}_a + \boldsymbol{\chi}_a^T\boldsymbol{\chi}_a) \triangleq -M
\end{aligned} \tag{49}$$

where $\boldsymbol{Z} = \left[(\boldsymbol{z}^T, \boldsymbol{\eta}_a^T, \boldsymbol{\chi}_a^T\right]^T$, $\boldsymbol{z} = [|z_1|, |z_2|, |z_3|]^T$, $\boldsymbol{\eta}_a = [|\eta_1|, |\eta_2|, |\eta_3|]^T$, $\boldsymbol{\chi}_a = [|\chi_1|, |\chi_2|]^T$, and $\lambda_{\min}(\boldsymbol{\Lambda})$ denotes the minimum eigenvalue of the matrix $\boldsymbol{\Lambda}$.

Hence, $V_1 \in L_\infty$, $M \in L_2$, and the boundness of all the system's signals holds. Based on error dynamics, the time derivative of $M$ is bounded as well. Thus, it is concluded that $M$ is uniformly continuous. Using the Barbalat's lemma [31], $M \to 0$ as $t \to \infty$. Consequently, the asymptotic stability of the closed-loop system can be ensured, i.e., the tracking error converges to zero asymptotically. As a result, Theorem 1 holds. $\quad\square$

**Theorem 2.** *If the unknown nonlinearities are time-varying (i.e., $H_1(t) \neq 0$ and $H_2(t) \neq 0$), with Assumptions 1–3 and the control law (40), by picking up the proper control parameters, $k_1$, $k_2$, and $k_3$, then it can be concluded that all the closed-loop signals are uniformly bounded.*

**Proof of Theorem 2.** For this case, define $x_{e1} = \Delta_1(t) - \widetilde{\boldsymbol{\theta}}^T\boldsymbol{\varphi}_2$ and $x_{e2} = \overline{\Delta}_2(t) - \widetilde{\boldsymbol{\theta}}^T\boldsymbol{\varphi}_3$, and choose a Lyapunov function as:

$$V_2 = \frac{1}{2}z_1^2 + \frac{1}{2}a_2z_2^2 + \frac{1}{2}a_3z_3^2 + \frac{1}{2}c_2\boldsymbol{\eta}^T\boldsymbol{P}_1\boldsymbol{\eta} + \frac{1}{2}c_3\boldsymbol{\chi}^T\boldsymbol{P}_2\boldsymbol{\chi} \tag{50}$$

According to (32), (34), (37) and (41), there is:

$$\begin{aligned}
\dot{V}_2 &= z_1(z_2 - k_1z_1) + a_2z_2(\frac{\partial x_p}{\partial x_1}z_3 - k_2z_2 + \widetilde{x}_{e1}) + a_3z_3(-k_3z_3 + \widetilde{x}_{e2} - \frac{\partial \alpha_2}{\partial x_2}\widetilde{x}_{e1}) \\
&\quad - \frac{1}{2}c_2\omega_{o1}\|\boldsymbol{\eta}\|^2 + c_2\boldsymbol{\eta}^T\boldsymbol{P}_1\boldsymbol{B}_2\frac{H_1(t)}{\omega_{o1}^2} - \frac{1}{2}c_3\omega_{o2}\|\boldsymbol{\chi}\|^2 + c_3\boldsymbol{\chi}^T\boldsymbol{P}_2\boldsymbol{B}_4\frac{H_2(t)}{\omega_{o2}}
\end{aligned} \tag{51}$$

One has:

$$
\begin{aligned}
\dot{V}_2 \leq\ & -k_1 z_1^2 + |z_1||z_2| + a_2 \frac{\partial x_p}{\partial x_1}|z_2||z_3| - k_2 a_2 z_2^2 + a_2 \omega_{o1}^2 |z_2||\eta_3| \\
& -k_3 a_3 z_3^2 + a_3 \omega_{o2}|z_3||\chi_2| + a_3 \omega_{o1}^2 \left|\frac{\partial \alpha_2}{\partial x_2}\right||z_3||\eta_3| - \tfrac{1}{2}c_2(\omega_{o1}-1)\|\eta\|^2 \\
& -\tfrac{1}{2}c_2\|\eta\|^2 + c_2\|\eta\|\frac{\|P_1 B_2\|\|H_1(t)|_{\max}}{\omega_{o1}^2} - \tfrac{1}{2}c_3(\omega_{o2}-1)\|\chi\|^2 - \tfrac{1}{2}c_3\|\chi\|^2 \\
& +c_3\|\chi\|\frac{\|P_2 B_4\|\|H_2(t)|_{\max}}{\omega_{o2}} \\
\leq\ & -Z^T \Lambda Z + \zeta
\end{aligned}
\tag{52}
$$

in which

$$
\zeta = \frac{1}{2}c_2\left(\frac{\|P_1 B_1\|\|H_1(t)|_{\max}}{\omega_{o1}^2}\right)^2 + \frac{1}{2}c_3\left(\frac{\|P_2 B_4\|\|H_2(t)|_{\max}}{\omega_{o2}}\right)^2
\tag{53}
$$

This results in:

$$
\begin{aligned}
\dot{V}_2 \leq\ & -\lambda_{\min}(\Lambda)(\|z\|_2 + \|\eta\|_2 + \|\chi\|_2) + \zeta \\
\leq\ & -\lambda_{\min}(\Lambda)\left[\lambda_1(z_1^2 + a_2 z_2^2 + a_3 z_3^2) + \frac{1}{c_2\lambda_{\max}(P_1)}c_2\eta^T P_1 \eta + \frac{1}{c_3\lambda_{\max}(P_2)}c_3\chi^T P_2 \chi\right] + \zeta \\
\leq\ & -\lambda_2 V_2 + \zeta
\end{aligned}
\tag{54}
$$

where $\lambda_1 = min\{1, 1/a_2, 1/a_3\}$, $\lambda_2 = 2\lambda_{min}(\Lambda) min\{\lambda_1, 1/(c_2\lambda_{max}(P_1)), 1/(c_3\lambda_{max}(P_2))\}$, and $\lambda_{\max}(\cdot)$ and $\lambda_{\min}(\cdot)$ denote the maximum and minimum eigenvalues of the matrix $\cdot$, respectively.

Hence, using the comparison Lemma [31], it is easy to obtain:

$$
V_2(t) \leq V_2(0)e^{-\lambda_2 t} + \frac{\zeta}{\lambda_2}(1 - e^{-\lambda_2 t})
\tag{55}
$$

Consequently, the uniformly ultimately bounded stability of the closed-loop system can be guaranteed, i.e., the tracking error is bounded, and all the system's signals are bounded. Thus, Theorem 2 holds. □

## 4. Simulation Results

The physical parameters of the hydraulic lifting system are collected in Table 1. The function was set as $S_f(x_2) = 2\arctan(900x_2)/\pi$, and the disturbances were set as $d_1(t) = 50000sin(t)\,N\cdot m$, $d_{21}(t) = 5 \times 10^{-5} \sin(\pi t)\,\text{m}^3/\text{s}$, $d_{22}(t) = -5 \times 10^{-5} \sin(\pi t)\,\text{m}^3/\text{s}$. The valve dead-zone nonlinear function was set as:

$$
x_v(u) = DZ(u) = \begin{cases} u - 1, & \text{if } u \geq 1 \\ 0, & \text{if } -1 < u < 1 \\ u + 1, & \text{if } u \leq -1 \end{cases}
\tag{56}
$$

where $m_r = 1$, $m_l = 1$, $b_r = 1$, and $b_l = 1$. The sample time was set as 0.5 ms.

**Table 1.** Physical parameters of the hydraulic lifting system.

| Parameter | Value | Parameter | Value |
|---|---|---|---|
| $m$ (kg) | 10,000 | $B(\text{N}\cdot\text{m}\cdot\text{s}/\text{rad})$ | $2.5 \times 10^5$ |
| $J$ (kg·m²) | $1.5 \times 10^5$ | $A_f$ (N·m) | $3 \times 10^3$ |
| $P_s$ (Pa) | $2.1 \times 10^6$ | $k_t\left(\text{m}^4/(\text{s}\cdot\text{V}\cdot\sqrt{\text{N}})\right)$ | $7.937\times 10^{-8}$ |
| $P_r$ (Pa) | 0 | $\beta_e$ (Pa) | $7 \times 10^8$ |
| $A_1$ (m²) | $3.14 \times 10^{-2}$ | $C_t\ (\text{m}^5/(\text{s}\cdot\text{N}))$ | $9.6 \times 10^{-13}$ |
| $A_2$ (m²) | $1.6 \times 10^{-2}$ | $L_1$ (m) | 1.6 |
| $V_{01}$ (m³) | $3.1416 \times 10^{-4}$ | $L_2$ (m) | 2 |
| $V_{02}$ (m³) | $3.04 \times 10^{-2}$ | $L_3$ (m) | 3.5 |
| $g$ (m/s²) | 9.8 | $L_4$ (m) | 3 |
| $q_0$ (rad) | 0.2648 | $\beta_0$ (rad) | 0.2618 |

Three controllers were compared to verify the validity of the developed controller.

(1) ADRAC: This controller was introduced in Section 3, and the controller parameters were provided by $k_1 = 200$, $k_2 = 50$, $k_3 = 30$, $\mathbf{\Gamma}$ = diag $\{1.1 \times 10^{-4}, 10.5, 4 \times 10^{-3}, 2.6 \times 10^{-9}, 1.8 \times 10^6, 3.5 \times 10^{-7}\}$, $a_2 = 1$, $a_3 = 0.01$, $c_2 = 0.01$, $c_3 = 0.01$, $\omega_{o1} = 200$, $\omega_{o2} = 200$, $\boldsymbol{\theta}_{\max} = [1, 10, 1, 1 \times 10^{-2}, 2 \times 10^5, 1 \times 10^{-7}]^{\mathrm{T}}$, and $\boldsymbol{\theta}_{\min} = [0, 0, 0, 0, 1 \times 10^2, -1 \times 10^{-7}]^{\mathrm{T}}$. The initial parameter estimation values were set as $\hat{\boldsymbol{\theta}}_0 = [0.05, 0, 0, 4 \times 10^{-4}, 3.5 \times 10^3, 4 \times 10^{-9}]^T$.

(2) ADRC: This is an active disturbance rejection control without parameter adaption. The difference between the ADRC and the ADRAC was that the parameter adaption matrix was set as $\mathbf{\Gamma} = 0$ in the ADRC. The other control parameters were the same as the ADRAC.

(3) AC: This is an adaptive controller without disturbance compensation. The difference between the AC and the ADRAC was that the observer parameters were set as $\omega_{o1} = 0$ and $\omega_{o2} = 0$ in AC. The other control parameters were the same as the ADRAC.

Case 1: First, a designed tracking trajectory, which had a maximum value of 45°, with maximum values of velocity and acceleration at the most at 2°/s and 2°/s², respectively, was employed to test the control performance of the developed control method. The simulation results are shown in Figures 3–7. Figure 3 shows the pitch angle tracking performance and tracking error under the ADRAC controller. It can be seen that although the overall steady-state tracking error was small (i.e., the actual pitch angle could well track the desired trajectory), there was an error jump due to the non-smooth characteristics of the dead-zone break-point of the valve. This shows that the error jump caused by the non-smooth valve dead-zone could not be eliminated, although the disturbance observer-based controller could ensure the overall tracking performance. The comparison of tracking errors between the ADRAC and the other two controllers is shown in Figure 4. By comparing the tracking errors of the ADRC and the ADRAC, the transient and steady-state tracking errors were both larger than those of the ADRAC, which indicates that the adaptive law adopted in the ADRAC could learn the part of the parameter uncertainties under the same observer bandwidth, thus reducing the burden of the extended state disturbance observer and achieving better control performance. The AC controller achieved the worst tracking performance because its parameter adaptation could not achieve good convergence under the influence of uncertain nonlinearities, which also verifies the validity of the disturbance compensation technique based on the extended state disturbance observer in the ADRAC. The parameter estimations of the system under the ADRAC controller are shown in Figure 5, all of which obtained good convergence characteristics. In addition, the estimations of mismatched and matched disturbances by the extended state observer are shown in Figure 6, and the control input voltage applied to the valve with the proposed ADRAC is presented in Figure 7.

Case 2: To further test the control performance of the developed control method, a new reference tracking signal, presented in Figure 8, was adopted. Similarly, it can be seen from Figures 8 and 9 that the ADRAC designed in this section still ensured the best tracking performance, which further verifies the validity of its adaptive law and extended state disturbance observer design. The parameter estimations, disturbance estimations, and control input of the ADRAC are shown in Figures 10–14, respectively. Consequently, the merits of the presented control method are verified by Case 2 once again.

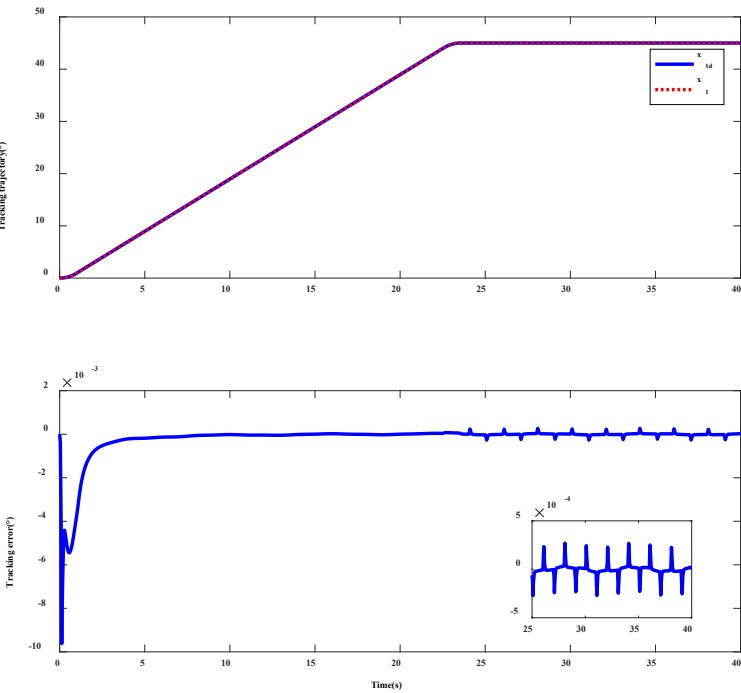

**Figure 3.** The tracking performance of the ADRAC.

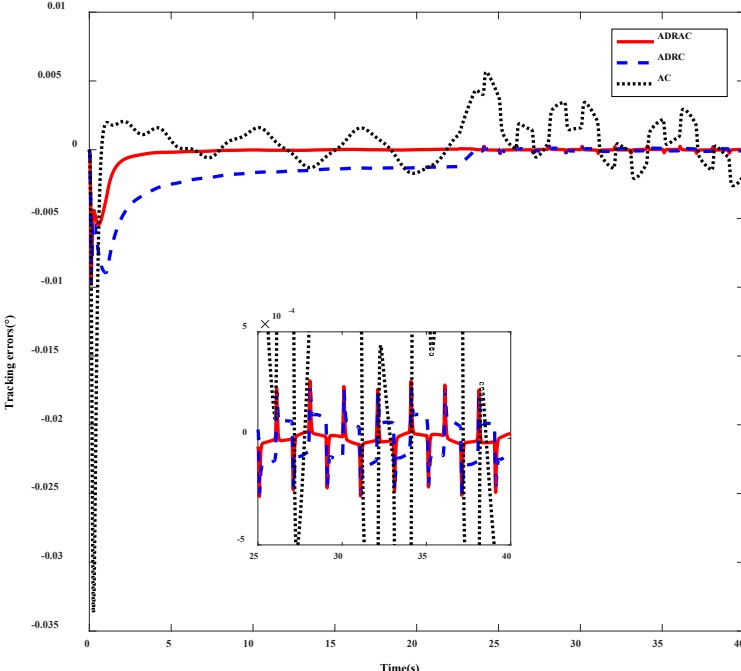

**Figure 4.** The tracking errors of the three controllers.

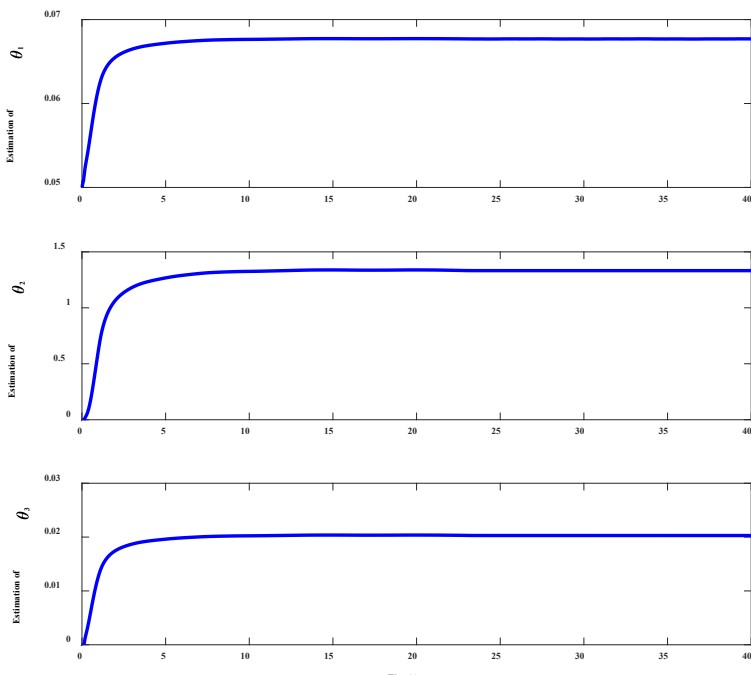

**Figure 5.** The parameter estimations of $\theta_1 \sim \theta_3$ with the ADRAC.

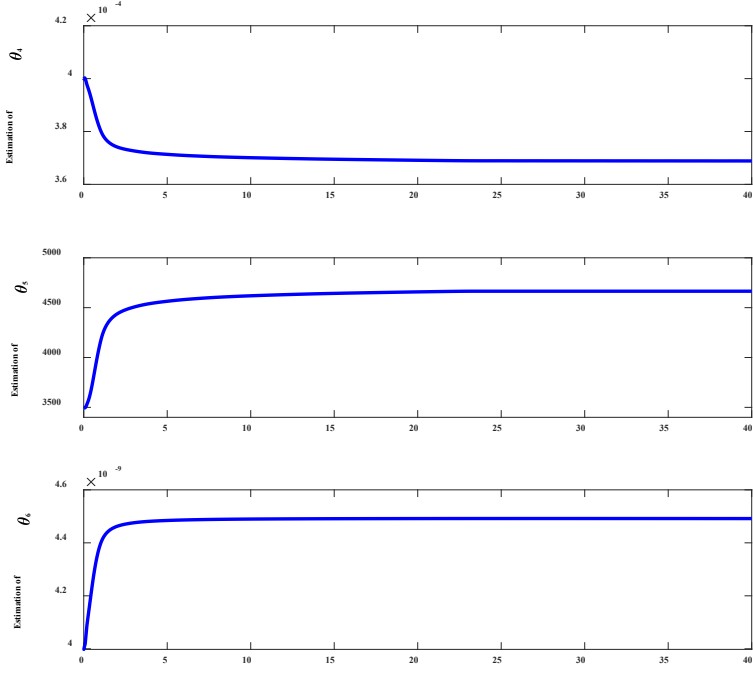

**Figure 6.** The parameter estimations of $\theta_4 \sim \theta_6$ with the ADRAC.

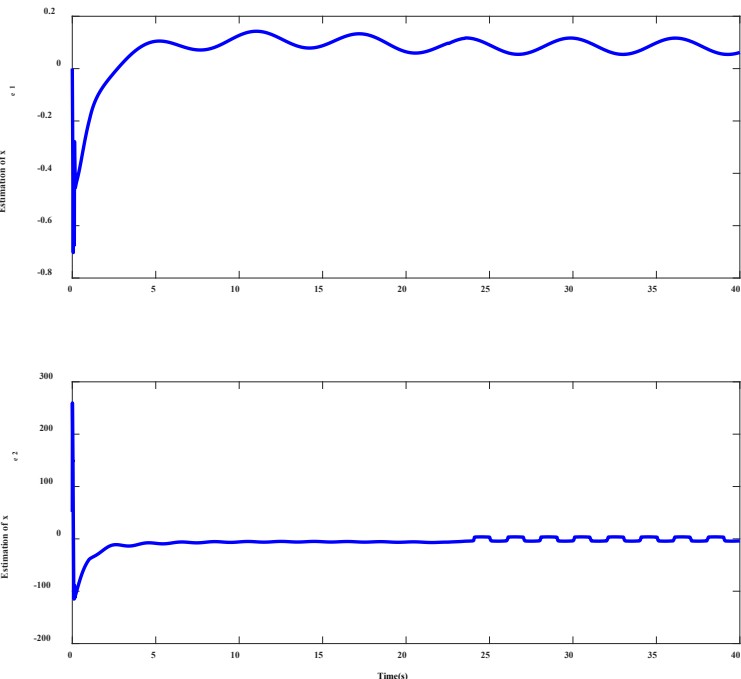

**Figure 7.** The disturbance estimations of the ADRAC.

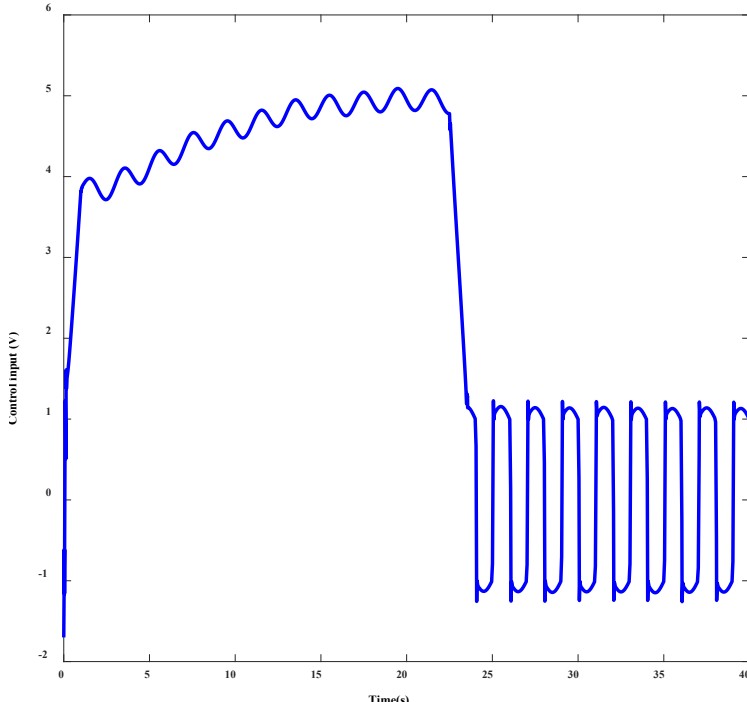

**Figure 8.** The control input of the ADRAC.

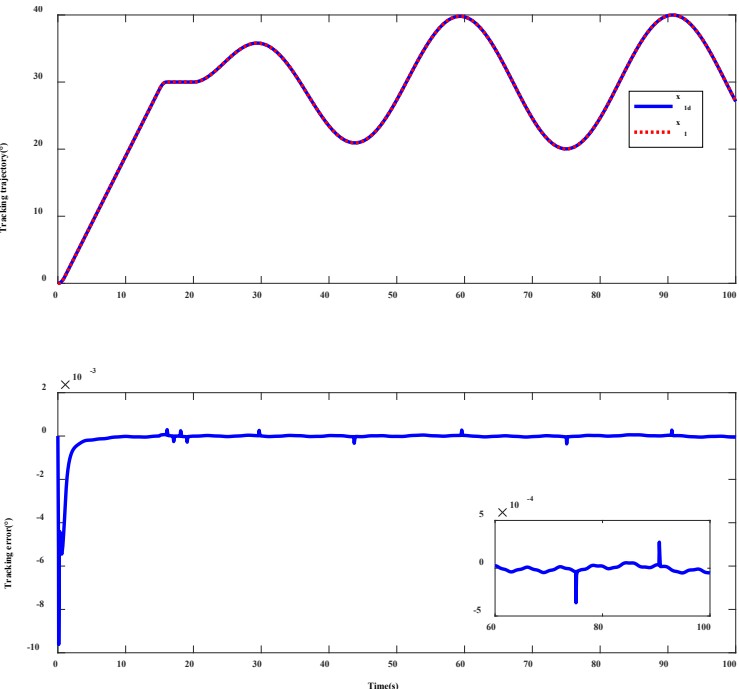

**Figure 9.** The tracking performance of the ADRAC.

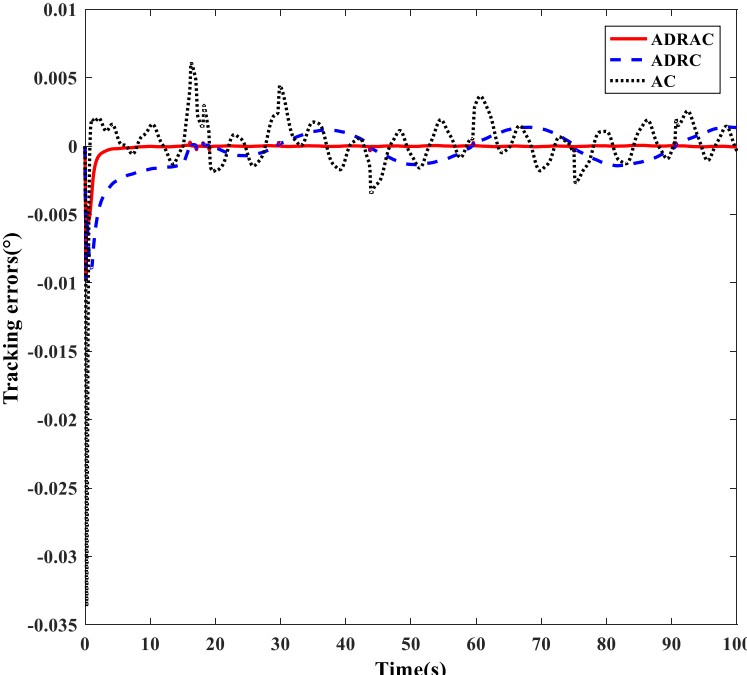

**Figure 10.** The tracking errors of the three controllers.

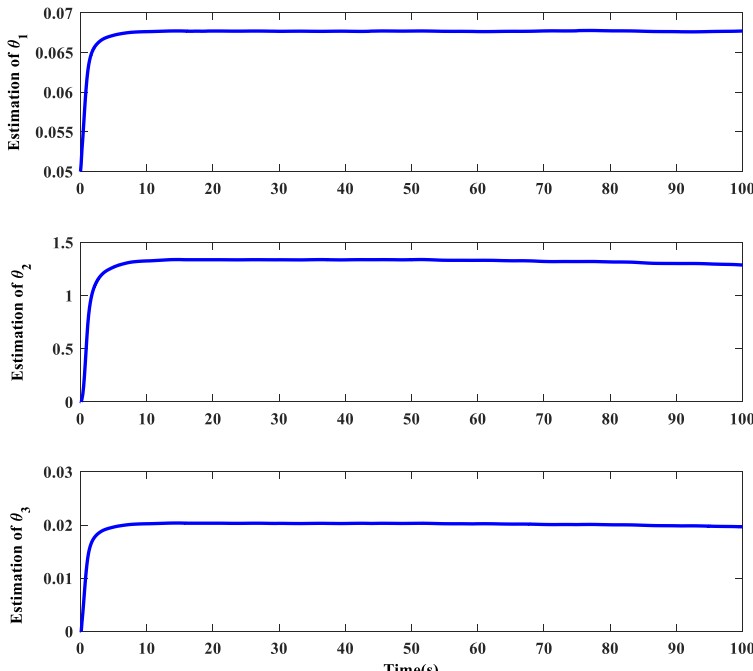

**Figure 11.** The parameter estimations of $\theta_1 \sim \theta_3$ with the ADRAC.

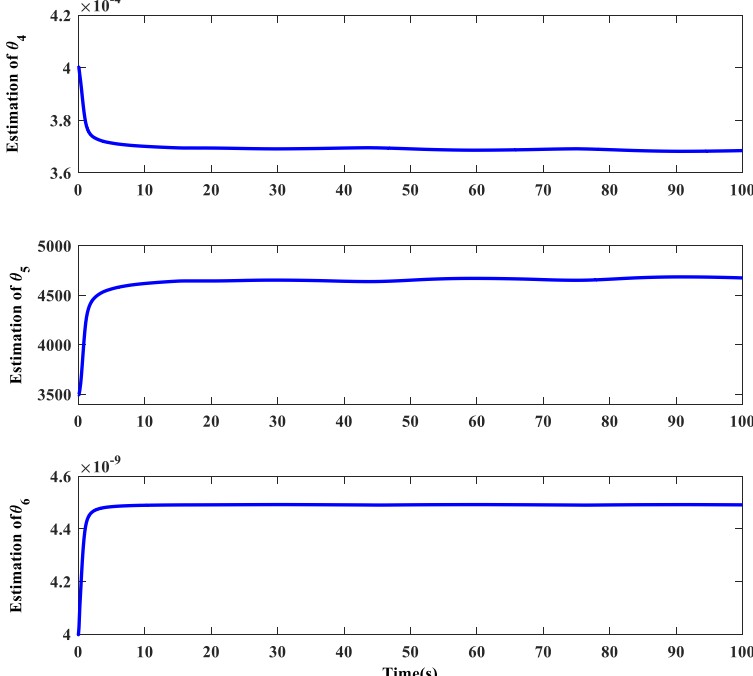

**Figure 12.** The parameter estimations of $\theta_4 \sim \theta_6$ with the ADRAC.

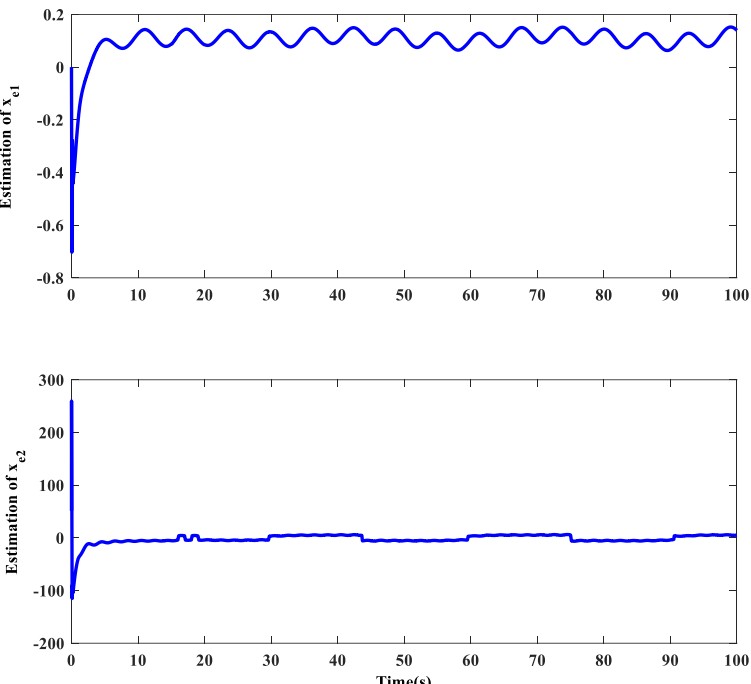

**Figure 13.** The disturbance estimations of the ADRAC.

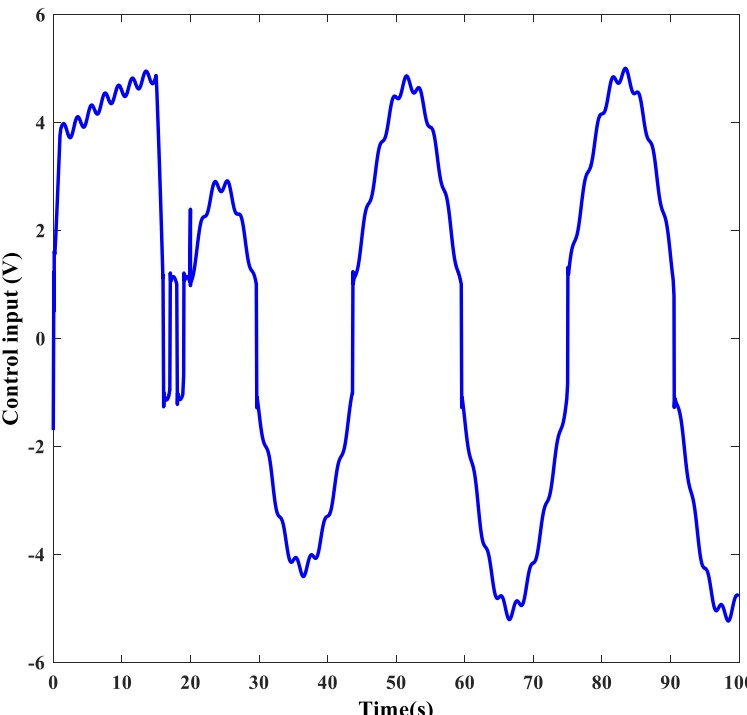

**Figure 14.** The control input of the ADRAC.

## 5. Conclusions

In this paper, an active disturbance rejection adaptive controller was developed for hydraulic lifting systems subject to parametric uncertainties, unmodeled disturbances, and a valve dead-zone. First, the dynamics, including both mechanical dynamics and hydraulic actuator dynamics with a valve dead-zone of the hydraulic lifting system, were modeled. Then, using the system model and backstepping technique, a composite parameter adaptation law and extended state disturbance observer were successfully combined, which were

used to dispose of the parametric uncertainties and unmodeled disturbances, respectively. This preserved the merits of both the control methods while surmounting their practical limitations. By using the Lyapunov function, the stability of the closed-loop system was assured. A simulation example of a hydraulic lifting system was performed to verify the validity of the proposed controller.

**Author Contributions:** Formal analysis, F.Y.; Investigation, W.D.; Methodology, F.Y.; Project administration, F.Y. and H.Z.; Writing—original draft, F.Y. All authors have read and agreed to the published version of the manuscript.

**Funding:** This research was supported in part by the National Natural Science Foundation of China (51705264 and 51905271) and in part by the Natural Science Foundation of Jiangsu Province (BK20190459).

**Conflicts of Interest:** The authors declare no conflict of interest.

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
