# Peer review of "Active Disturbance Rejection Adaptive Control for Hydraulic Lifting Systems with Valve Dead-Zone"

_electronics, doi:10.3390/electronics11111788_

Round 1

Reviewer 1 Report

The authors studied the motion control problem of hydraulic lifting systems subject to parametric uncertainties, unmodeled disturbances, and valve dead-zone. Both mechanical dynamics and hydraulic actuator dynamics with valve dead-zone of the hydraulic lifting system are modeled in detail. They also did stability analysis of the system. The theoretical results are verified with simulation results.

Comments:

1. Why the authors choose these values of Physical parameters of the hydraulic lifting system (Table 1).

2.      2. Add physical interpretation for obtained results in possible.

Reviewer 2 Report

This is a well written paper with appropriate and well-described and validated novelties. Before the publication, one thing should be clarified -- that this reviewer missed to understand:

- The authors claim that "the stability of the closed-loop system is assured". While Lyapunov functions are clearly described in the paper,  the process of validating the stability of the system is not. Please expand on this.

Reviewer 3 Report

The problem solved in the article is relevant and of interest to the scientific community. The statement of the problem and the methods used are correct.  The research results are reliable and trustworthy. The article should be recommended for publication.

Author Response

Thank your for your affirmative comments.